# Efficacy and Safety of Tinzaparin in Prophylactic, Intermediate and Therapeutic Doses in Non-Critically Ill Patients Hospitalized with COVID-19: The PROTHROMCOVID Randomized Controlled Trial

**DOI:** 10.3390/jcm11195632

**Published:** 2022-09-24

**Authors:** Nuria Muñoz-Rivas, Jesús Aibar, Cristina Gabara-Xancó, Ángela Trueba-Vicente, Ana Urbelz-Pérez, Vicente Gómez-Del Olmo, Pablo Demelo-Rodríguez, Alberto Rivera-Gallego, Pau Bosch-Nicolau, Montserrat Perez-Pinar, Mónica Rios-Prego, Olga Madridano-Cobo, Laura Ramos-Alonso, Jesús Alonso-Carrillo, Iria Francisco-Albelsa, Edelmira Martí-Saez, Ana Maestre-Peiró, Manuel Méndez-Bailón, José Ángel Hernández-Rivas, Juan Torres-Macho

**Affiliations:** 1Department of Internal Medicine, Hospital Universitario Infanta Leonor, 28031 Madrid, Spain; 2Universidad Complutense, 28040 Madrid, Spain; 3Department of Internal Medicine, Hospital Clínic, IDIBAPS, 08036 Barcelona, Spain; 4University of Barcelona, 08007 Barcelona, Spain; 5Department of Internal Medicine, Hospital de Emergencias Enfermera Isabel Zendal, 28055 Madrid, Spain; 6Department of Internal Medicine, Hospital Universitario Ramón y Cajal, 28034 Madrid, Spain; 7Department of Internal Medicine, Hospital General Universitario Gregorio Marañón, 28007 Madrid, Spain; 8Department of Internal Medicine, Hospital Álvaro Cunqueiro, 36312 Vigo, Spain; 9Department of Infectious Diseases, Hospital Universitario Vall d’Hebron, 08035 Barcelona, Spain; 10Department of Internal Medicine, Hospital Virgen de la Luz, 16002 Cuenca, Spain; 11Department of Internal Medicine, Complexo Hospitalario Universitario de Pontevedra, 36071 Pontevedra, Spain; 12Department of Internal Medicine, Hospital Universitario Infanta Sofía, 28702 San Sebastián de los Reyes, Spain; 13Department of Internal Medicine, Hospital Universitario A Coruña, 15006 A Coruña, Spain; 14Department of Internal Medicine, Hospital Universitario 12 de Octubre, 28041 Madrid, Spain; 15Department of Internal Medicine, Hospital Universitari de Girona Dr. Josep Trueta, 17007 Girona, Spain; 16Department of Haematology, Hospital Clínico Universitario de Valencia, 46010 Valencia, Spain; 17Department of Internal Medicine, Hospital Universitario de Vinalopó, 03293 Elche, Spain; 18Department of Internal Medicine, Hospital Clinico San Carlos, 28040 Madrid, Spain; 19Department of Hematology, Hospital Universitario Infanta Leonor, 28031 Madrid, Spain

**Keywords:** COVID-19, pulmonary embolism, thrombosis, respiratory insufficiency, low-molecular-weight heparin

## Abstract

Hospitalized patients with COVID-19 are at increased risk of thrombosis, acute respiratory distress syndrome and death. The optimal dosage of thromboprophylaxis is unknown. The aim was to evaluate the efficacy and safety of tinzaparin in prophylactic, intermediate, and therapeutic doses in non-critical patients admitted for COVID-19 pneumonia. PROTHROMCOVID is a randomized, unblinded, controlled, multicenter trial enrolling non-critical, hospitalized adult patients with COVID-19 pneumonia. Patients were randomized to prophylactic (4500 IU), intermediate (100 IU/kg), or therapeutic (175 IU/kg) groups. All tinzaparin doses were administered once daily during hospitalization, followed by 7 days of prophylactic tinzaparin at discharge. The primary efficacy outcome was a composite endpoint of symptomatic systemic thrombotic events, need for invasive or non-invasive mechanical ventilation, or death within 30 days. The main safety outcome was major bleeding at 30 days. Of the 311 subjects randomized, 300 were included in the prespecified interim analysis (mean [SD] age, 56.7 [14.6] years; males, 182 [60.7%]). The composite endpoint at 30 days from randomization occurred in 58 patients (19.3%) of the total population; 19 (17.1 %) in the prophylactic group, 20 (22.1%) in the intermediate group, and 19 (18.5%) in the therapeutic dose group (*p* = 0.72). No major bleeding event was reported; non-major bleeding was observed in 3.7% of patients, with no intergroup differences. Due to these results and the futility analysis, the trial was stopped. In non-critically ill COVID-19 patients, intermediate or full-dose tinzaparin compared to standard prophylactic doses did not appear to affect the risk of thrombotic event, non-invasive ventilation, or mechanical ventilation or death. **Trial Registration**
**ClinicalTrials.gov Identifier (NCT04730856).** Edura-CT registration number: 2020-004279-42.

## 1. Introduction

Severe acute respiratory syndrome-coronavirus 2 infection can cause different clinical manifestations, ranging from mild to very severe symptomatology, with significant morbidity and mortality, principally associated with bilateral pneumonia that can cause acute respiratory distress syndrome (ARDS). More than 6 million people have died since the first reports in late 2019 in Wuhan, China, and it is estimated that more than 450 million people have been infected with COVID-19 to date [1,2]. Recent research estimates that more than 18 million people have died worldwide because of the COVID-19 pandemic (as measured by excess mortality) over that period [1]. Since the first wave of the COVID-19 pandemic, the increase in systemic thrombosis in hospitalized patients was evident [2], particularly in critical care units worldwide [3,4]. The phenomenon known as ‘pulmonary immunothrombosis’ is correlated with the severity of respiratory failure and need for mechanical ventilation in individuals with COVID-19 [5]. The association of viral infection with thrombosis is mediated by two interrelated processes: a state of hypercoagulability that causes large vessel thrombosis and direct endothelial damage that provokes in situ thrombosis [5]. Subsequently, more and more evidence has been published of the so-called ‘COVID-19-associated coagulopathy’. It was then hypothesized that anticoagulation could improve clinical outcome of patients with COVID-19 infection who, given the severity of their disease, required hospitalization [5]. At the beginning of the pandemic, while awaiting the results of clinical trials, different protocols of prophylactic anticoagulation have been developed in hospitals. These included the use of standard, intermediate, and even full doses of low-molecular-weight heparin (LMWH) [6]. This was the underlying premise for conducting numerous clinical studies to evaluate the efficacy and safety of therapeutic or intermediate doses (with either LMWH or different oral anticoagulants) versus prophylactic doses of anticoagulation. The results of several clinical trials have been published to date, focusing on anticoagulation intensity in patients admitted for COVID-19 [7]. Uncertainty persists as to the optimal LMWH doses in non-critical cases [8,9]. Most trials have evaluated standard prophylactic LMWH dose strategies versus therapeutic doses or other oral anticoagulants, with contradictory results [10,11].

The PROTHROMCOVID multicenter clinical trial was conducted to evaluate the efficacy of tinzaparin treatment at different doses (prophylactic, intermediate, and therapeutic) in patients with COVID-19 non-critical pneumonia to probe the endpoints of death, need for mechanical ventilation and venous or arterial thrombosis within 30 days following randomization. This trial also examined the safety of tinzaparin at different doses in relation to the risk of both major and minor bleeding.

## 2. Materials and Methods

### 2.1. Study Design

The PROTHROMCOVID study (NCT04730856) is a randomized, open-label, unblinded, multicenter, controlled study in hospitalized patients with COVID-19 pneumonia (defined by consolidations/infiltrations on chest X-ray or CT scans), conducted in conventional hospital wards in 18 academic hospitals in Spain. This investigator-initiated clinical trial enrolled individuals with COVID-19 pneumonia who were hospitalized from 1 February 2021 to 30 September 2021. The trial follows the CONSORT guideline as detailed by EQUATOR network. Edura-CT registration number: 2020-004279-42.

### 2.2. Patients

Adults with a body weight of 50–100 kg who required admission to a conventional (non-critical) hospital ward due to COVID-19 pneumonia were included if they also met any of the following criteria: (a) baseline oxygen saturation ≤ 94%, (b) D-dimer > 1000 µg/L, (c) C Reactive Protein (CRP) > 150 mg/L, or (d) interleukin-6 (IL6) > 40 pg/mL. The main exclusion criteria were: (a) the need for full-dose anticoagulant therapy, (b) active bleeding or situations prone to bleeding, (c) glomerular filtration rate < 30 mL/min/1.73 m^2^, (d) platelet count < 80 × 10^9^/L, (e) previous heparin-induced thrombocytopenia, and (f) hypersensitivity/intolerance to heparins. The study design (Figure 1) and full list of eligibility and exclusion criteria are listed below:


**Inclusion Criteria:**
Patients admitted for COVID-19 pneumonia;Patients with at least one of the following risk criteria for disease progression:-Sat 02 < 94%-DD > 1000 µg/L-CRP > 150 mg/L-IL6 > 40 pg/mL;Age > 18 years;Weight between 50 and 100 kg;After receiving verbal and written information about the study, the patient must submit the signed and dated Informed Consent before carrying out any activity related to the study.



**Exclusion Criteria:**
Patients who require mechanical ventilation or ICU admission at the time of randomization;Current diagnosis of acute bronchial asthma attack;History or clinical suspicion of pulmonary fibrosis;Current diagnosis of suspected pulmonary thromboembolism;Patients who require anticoagulant or antiplatelet therapy for a previous venous or arterial thrombotic disease;Patients with pneumonectomy or lobectomy;Kidney failure with GFR <30 mL/min;Patients with contraindication to anticoagulation;Congenital bleeding disorders;Hypersensitivity to tinzaparin or HNF or to any of its excipients;History of heparin-induced thrombocytopenia;Active bleeding or situations that predispose to bleeding;Moderate or severe anemia (Hb < 10 g/dL);Platelet count < 80,000/µL;Patients with a life expectancy of less than 3 months due to the primary disease evaluated by the physician.


### 2.3. Randomization

Patients were screened on admission and randomized at a ratio of 1:1:1 by means of a central, electronic, automated system with permuted blocks of 6. Neither participants, nor investigators were blinded as to group assignment. Subjects were stratified by age, sex, and presence of high blood pressure. Those who were assigned to the control group received standard prophylaxis with subcutaneous (sc) tinzaparin 4500 IU once daily. The experimental group received tinzaparin 100 IU/kg once daily (intermediate dose group) or 175 IU/kg once daily (therapeutic dose group) (Figure 1). The first dose of tinzaparin was administered within the first 24 h after randomization. Prior to randomization, patients could receive prophylactic or higher dose LWMH as local protocol of each center. Recommendations from the Spanish Society of Thrombosis and Hemostasis were followed by most centers and a dose-escalating protocol was implemented depending on risk and clinical severity/prognostic factors. The assigned treatments remained the same throughout hospitalization. Therapeutic-dose anticoagulation treatment was applied if patients developed a thromboembolic event, atrial fibrillation, or any clinical condition requiring anticoagulation according to clinical guidelines. After discharge, all patients received tinzaparin 4500 IU/day subcutaneously for seven days, after which thromboprophylaxis was maintained at the discretion of the attending physician. If intensive care unit (ICU) admission was required, the patients could remain with the study drug or not, according to local practices. Except for the assigned anticoagulation therapy, all other clinical care was provided as per local protocols.

### 2.4. Outcomes

Demographic characteristics, comorbidities, medications, and laboratory evaluations were recorded at randomization. The primary efficacy outcome was a composite endpoint of death, need for invasive mechanical ventilation (IMV), non-invasive ventilation (NIV), including high flow oxygen with nasal cannula (HFNC), and venous or arterial thrombosis within 30 days after randomization. Safety outcomes were major bleeding and clinically relevant non-major bleeding, as defined by the International Society on Thrombosis and Hemostasis (ISTH) [12]. Secondary outcomes of the trial were:Reduction of suspected systemic thrombotic events (myocardial infarction, ischemic stroke, deep vein thrombosis, pulmonary thromboembolism confirmed with imaging tests);Progression on the WHO progression scale (worst situation during admission and at discharge);Progression to Acute Respiratory Distress Syndrome by PaO_2_/FiO_2_ or SpO_2_/FiO_2_ criteria;Overall survival day 14, 30, and 90;Length of hospital stay;Orotracheal intubation;Length of ICU stay;Incidence of major bleeding;Incidence of clinically relevant non-major bleeding;Incidence of clinically relevant bleeding;Incidence of adverse reactions;Changes in biochemical and hematological values from Day 1 to Day 14 between groups.

Outcomes were adjudicated locally by one investigator based on objectively confirmed diagnostic tests, laboratory results, and other objective data from the clinical record. The diagnosis of thrombosis was based on clinical suspicion. DVT was defined as a non-compressible venous segment on ultrasonography and a PE was diagnosticated as intraluminal filling defect in the spiral CT or in the pulmonary angiography or signs suggestive of PE in the echocardiography.

### 2.5. Statistical Analysis

Considering the main objective, the incidence in the prophylactic group was expected to be 24% and 12% in intermediate group and therapeutic group-based internal hospital data from 1–12 April 2020.

The sample size was calculated from a proportion of a 13% reduction in thrombosis in the prophylactic group. It was assumed that with therapeutic doses we would be able to reduce the risk of thrombosis by 8% (from 13 to 5%). The remaining 4% reduction (from 11% to 7%) would be obtained from the reduction in the other component variables of the main variable (death, need for invasive mechanical ventilation or high-flow ventilation).

Accepting an alpha risk of 0.025 and a beta risk of <0.2 in a bilateral contrast, statistically significant differences could be detected with 200 patients per group. The study protocol included an interim analysis when 50% of the target population had been included. An interim analysis was scheduled to be performed after 300 patients were included. The trial could be stopped for: (1) superiority; (2) futility with regard to the primary endpoint; or (3) safety reasons. Following the results of this interim analysis presented in this article, the Scientific Committee decided to prematurely halt the clinical trial, based on the futility analysis and the drop in recruitment at the end of fifth wave.

Categorical variables were expressed as frequencies and percentages, and quantitative variables as mean ± standard deviation (SD) or median and interquartile range (IQR), relative to distribution. The Shapiro–Wilk test was used to examine the normality of the distributions of samples of <30 and the Kolmogorov–Smirnoff test was applied in the other cases. For intergroup statistical analysis, chi-square or Fisher’s exact test were used for categorical variables and unpaired Student’s t-test or Mann–Whitney test for continuous variables. Survival analysis was performed using Kaplan–Meier curves. Efficacy and safety were assessed in the modified intention-to-treat population, including all randomized patients who received at least one dose of the assigned treatment. Statistical analyses were performed using the statistical package SAS, 9.4 (Copyright © 2016 by SAS Institute Inc., Cary, NC, USA).

## 3. Results

From 1 February 2021 to 30 September 2021, 311 patients were enrolled, coinciding with the third to the fifth pandemic wave in Spain, 11 subjects were excluded from the analysis due to withdrawal of consent or screening failure, while all other patients did receive at least one dose and were all included in the analysis. Among these patients, the intention-to-treat, per-protocol, and safety populations were equally constituted, with no major protocol deviations detected and all treatment doses received. The study protocol included an interim analysis with 50% of the estimated sample size, at which point the Scientific Committee decided to discontinue the study in light of the results presented below.

Of the 300 patients, 106 (35.3%) were assigned to the prophylaxis group; 91 patients (30.3%) were allocated to the intermediate dose group; and 103 patients (34.3%) were randomized to the therapeutic dose group (flow chart in Figure 2).

Baseline characteristics were similar in the three groups, including D-dimer, IL6, CRP, and ferritin values (Table 1). IL6, CRP, and ferritin are presented as values upon hospital admission.

The distribution of individuals with D-dimer <1000 µg/L was as follows: 83% in the prophylaxis group, 72% in the intermediate dose group, and 79% in the therapeutic dose group. Treatment for COVID-19 with corticosteroids (89.3%), remdesivir (18.0%), or tocilizumab (14.3%) was comparable in all three groups. The percentage of COVID-19-vaccinated subjects was 16%, 29%, and 26% in the prophylaxis, intermediate, and therapeutic dose groups of tinzaparin, respectively (*p* = 0.06). There was one oncological patient in each group and no patients had hematological diseases.

Primary endpoint: The composite endpoint, which ensued in 58 participants (19.3%) of the total study population: 19 patients (17.9%) in the prophylactic dose group, 20 (22.0%) in the intermediate dose group, and 19 (18.4%) in the therapeutic dose group (*p* = 0.72). (Table 2).

The survival analysis revealed no statistically significant intergroup differences at 30 days. Prophylactic dose group, 0.82 CI: 95% (0.73–0.88); intermediate dose group, 0.78 CI: 95% (0.68–0.85); therapeutic dose group: 0.81 CI: 95% (0.73–0.88); Log-rank test *p*-value = 0.75) (Figure 3).

No differences were observed in survival when the groups were stratified according to D-dimer values (*p* = 0.40) (Appendix C, Figure A3) or between vaccinated or non-vaccinated patients: 29.31% vs. 25.52% (*p* = 0.55). 

In terms of safety, the rate of bleeding was very low in all three groups. No major bleeding was reported and seven patients (6.6%) in the prophylactic dose group, three participants (3.2%) in the intermediate group, and three patients (2.9%) in the therapeutic dose group suffered non-major bleeding, with no significant differences across groups (*p* = 0.38).

A thrombotic event occurred in four patients in the prophylaxis group (3.8%); in two patients (2.2%) in the intermediate dose group, and in two subjects (1.9%) in the therapeutic dose group. NIV was provided for 10.5% of the prophylactic dose group, 11.8% in the intermediate group, and 4.9% in the therapeutic group. Seven (2.3%) of the included participants died during the first 30 days; two in the prophylactic dose group, three in the intermediate dose group, and two in the therapeutic dose group (*p* = 0.48).

The World Health Organization (WHO) progression scale indicated no intergroup differences in progression between the date of admission and day 4, day 7, and at discharge. As for respiratory interventions, at day 4, 10% of the patients did not require oxygen therapy; 87% required oxygen therapy with nasal goggles or non-rebreather facemask; 0.74% required HFNC or NIV; 0.4% needed NIV, and 0.4% required IMV. The Wilcoxon paired signs test showed no differences in progression between groups.

The results of a per-protocol analysis were similar to the intention-to-treat analysis.

Futility analysis showed that there was no evidence of significant differences between the prophylactic dose group and the therapeutic dose group (Z = −0.09, *p* = 0.92); comparing prophylactic dose group and intermediate dose group, the results obtained were very close to entering the zone of non-rejection of the null hypothesis, (Z = −0.71, *p* = 0.48) and boundary values: α = −2.72; β = −0.70 (Appendix A, Appendix B).

## 4. Discussion

The results of the PROTHROMCOVID trial did not show differences during treatment with tinzaparin in relation to prophylactic, intermediate, or therapeutic doses in relation to the probability of death, thrombotic event, or non-invasive ventilation or invasive mechanical ventilation in patients with COVID-19 pneumonia. In this regard, the results of our trial provide evidence on the use of LMWH, indicating that there seems to be no advantage of a higher dose although the risk of major bleeding appears to be low regardless of dose in hospitalized and non-critical patients with pneumonia due to COVID-19. This study tested these three strategies of different LMWH doses that coexisted *de facto* in different hospitals in the absence of solid evidence of the most suitable dose and faced with the high rate the high rate of thrombosis and respiratory failure recorded in the first wave of the pandemic. 

The results of the PROTHROMCOVID trial are in line with a previous study published by Lópes et al. The ACTION trial, conducted at the end of first and second waves of the pandemic, included 615 patients and used a hierarchical statistical analysis structure based on time to death, and it detected no survival benefit or in duration of hospitalization in individuals treated with full-dose enoxaparin or rivaroxaban compared to those who received standard prophylactic LMWH doses [11].

Similarly to our results, the RAPID trial determined that there was no significant difference between therapeutic or prophylactic strategies in non-critically ill patients admitted for COVID-19 in the combined endpoint of death, mechanical ventilation, or ICU admission [13]. Moreover, the BEMICOP clinical trial, a small study conducted with bemiparin, did not find any differences in the primary endpoint between cases randomized to therapeutic doses in comparison with prophylactic doses [14]. In contrast, the results of REMAP-CAP, ACTIV-4a, and ATTAC multiplatform collaborative trials support an early strategy of full-dose anticoagulant doses of heparin in non-critically ill subjects by demonstrating an increased in organ-free support days evaluated on an ordinal scale that combined in-hospital death and the number of days free of cardiovascular or respiratory organ support up to day 21 among patients who survived to hospital discharge (98.6% vs. 95.0%, respectively) in comparison to standard doses of LMWH thromboprophylaxis [10]. Despite being the clinical trial that has included the largest number of patients, statistical significance was barely reached and there were no statistical differences in other outcomes among groups, including thrombosis, survival to hospital discharge, and bleeding. Moreover, the percentage of patients who received intermediate doses in the prophylaxis group was high (26%), which may have biased the results [15]. The HEPCOVID trial, with a dose design similar to PROTHROMCOVID trial, showed a decrease in events, thromboembolism, and death in the therapeutic-dose LMWH in hospitalized, but not in ICU patients, with no differences at the intermediate-dose level [16]. It should be noted that the HEPCOVID trial was conducted in May 2020, during the first wave, with a higher percentage of events than those observed in our trial and in those conducted in later stages of the pandemic. It is worth mentioning that PROTHROMCOVID recruitment began in February 2021, in the middle of the third wave of the pandemic in Spain and up to and including the fifth wave. Consequently, patients were at lower risk of mortality, given the widespread use of corticosteroids and the beginning of vaccination against COVID-19 (unlike other studies), with the first dose of tinzaparin administered within the first 24 h after randomization and with concomitant treatment, mainly corticosteroids, most of which were homogeneous across the patients included. This profile is more similar to current clinical presentations than those of the first wave of the pandemic.

In line with our results, two clinical trials have analyzed standard prophylactic versus intermediate-dose LMWH. The INSPIRATION trial [17] tested the effect of intermediate versus standard dose prophylactic anticoagulation on thrombotic events, extracorporeal membrane oxygenation treatment, or mortality among patients with COVID-19 admitted to ICU. Likewise, Perepu et al. [18] published the results of their trial that examined standard prophylactic versus intermediate-dose enoxaparin in adults with severe COVID-19; both trials did not find significant intergroup differences. The lack of efficacy of the intermediate or full doses compared to standard doses could be due to differences in the clinical situation of the subjects included after the first wave, in which patients displayed more inflammation and received fewer doses of corticosteroids, monoclonal antibodies, immunomodulators, and antivirals that have demonstrated benefit in the evolution of the disease. In addition, the severity of symptoms in individuals affected by COVID-19 variants with lower mortality rates in the last months of recruitment may account for these data. Similarly, the incidence of thrombosis recorded during the first wave [19] in our own setting was higher than data collected during the second wave.

A meta-analysis including 49 studies concluded that prophylactic anticoagulation was recommended rather than intermediate to therapeutic anticoagulation, considering insignificant survival benefits but higher risk of bleeding when higher doses were used [20]. The PROTHROMCOVID study confirms the non-superiority of intermediate doses and therapeutic doses with respect to standard prophylactic LMWH doses; consequently, the accumulated evidence suggests that this strategy should be abandoned in this patient group.

However, the recommendations of the different guidelines have not been unanimous either. The American Society of Hematology favored a prophylactic dose over intermediate or therapeutic dose for patients with critical illness related to COVID-19 or acute illness without confirmed or suspected thromboembolic disease [21], while the National Institute for Health and Care Excellence (NICE) guidelines put forth the conditional recommendation to consider a therapeutic dose of LMWH for young people and adults with COVID-19 who need low-flow oxygen and who do not have an increased bleeding risk [22].

In terms of safety, the risk of bleeding tends to be higher in most of the studies in which the anticoagulation strategy is more intense [23]. In multiplatform trials, the risk of major bleeding was 1.8% in controls receiving standard prophylaxis versus 3.7% in those receiving therapeutic doses [10]. The PROTHROMCOVID study participants had no major bleeding events, perhaps because of the smaller sample size than in the collaborative trials, the characteristics of the included population, or the type of heparin used [6].

We believe our safety data to be of the utmost importance because it does not appear from our results that the option of therapeutic anticoagulation or intermediate doses generates an increased risk of major bleeding in a subset of non-critically ill patients where upcoming ASH or ISTH guidelines may suggest full-dose LMWH as NICE guidelines does.

Our study has certain limitations. For instance, neither investigators, nor patients were blinded. The main weakness of our results, however, is not having reached the estimated sample size, given that the researchers chose to interrupt the study on September 2021 due to both the slow recruitment rate and the results of the interim analysis. After the first 300 patients, we conducted the planned interim analysis. At this point, there were 19 combined outcome events in the 103 patients who received standard prophylaxis tinzaparin 4500 IU/kg, 20 combined events in patients who received intermediate-dose tinzaparin 100 IU/kg/day, and 19 combined events in patients who received therapeutic-dose tinzaparin 175 IU/kg/day. It revealed a lower absolute number of events than expected, as well as a smaller relative difference between intermediate and therapeutic versus standard prophylaxis, so it was unlikely that significant differences could have been reached with the complete sample originally planned. We determined that we would need at least 2592 patients per group to achieve a statistically significant difference.

These results should not be extrapolated to other more severe hospitalized patients with COVID-19.

The strengths of our study include the low number of withdrawals of informed consent by patients, the very early use of tinzaparin in all three arms of the study, which may have influenced in the safety outcomes, and the fact that the three strategies of anticoagulation were with the same LMWH. In Spain, LMWH such as tinzaparin, enoxaparin, or bemiparin, among others, are approved for the prophylaxis and treatment of venous thromboembolic disease. We consider that very few results had been reported on the use of tinzaparin in the prophylaxis of thromboembolism associated with COVID-19. Therefore, we consider that this fact could provide more evidence in this field.

Similarly, this was a multicenter study conducted in academic and general care centers. Furthermore, the study was conducted during a phase of the pandemic in which the incidence of thrombosis and mortality were lower than before; thus, the findings of our study might be more applicable to future waves of the pandemic, which are expected to be milder due to generalized immunization, fewer pathogenic variants of COVID-19, and better treatment options [24].

## 5. Conclusions

In conclusion, in non-critically ill COVID-19 pneumonia patients, intermediate, or full-dose tinzaparin does not appear to offer any benefit over standard, prophylactic doses, on the risk of thrombotic events, use of invasive or non-invasive ventilation, high-flow oxygen with nasal cannula, or death. However, the risk of bleeding related to intermediate or full heparin doses appears to be low in these patients.

PROTHROMCOVID trial (NCT04730856).

## Figures and Tables

**Figure 1 jcm-11-05632-f001:**
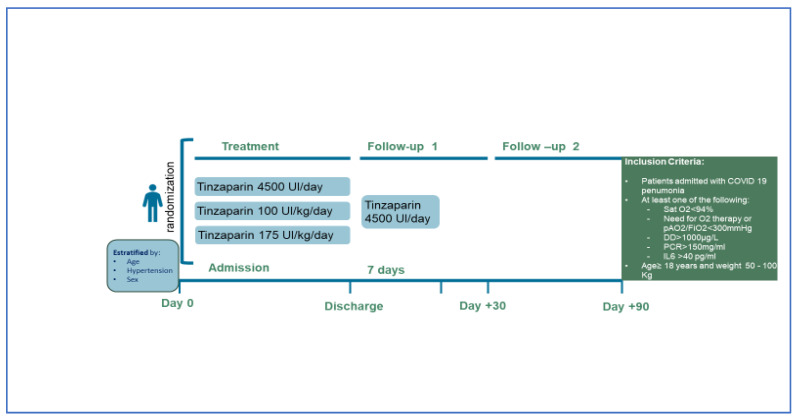
PROTHROMCOVID trial design.

**Figure 2 jcm-11-05632-f002:**
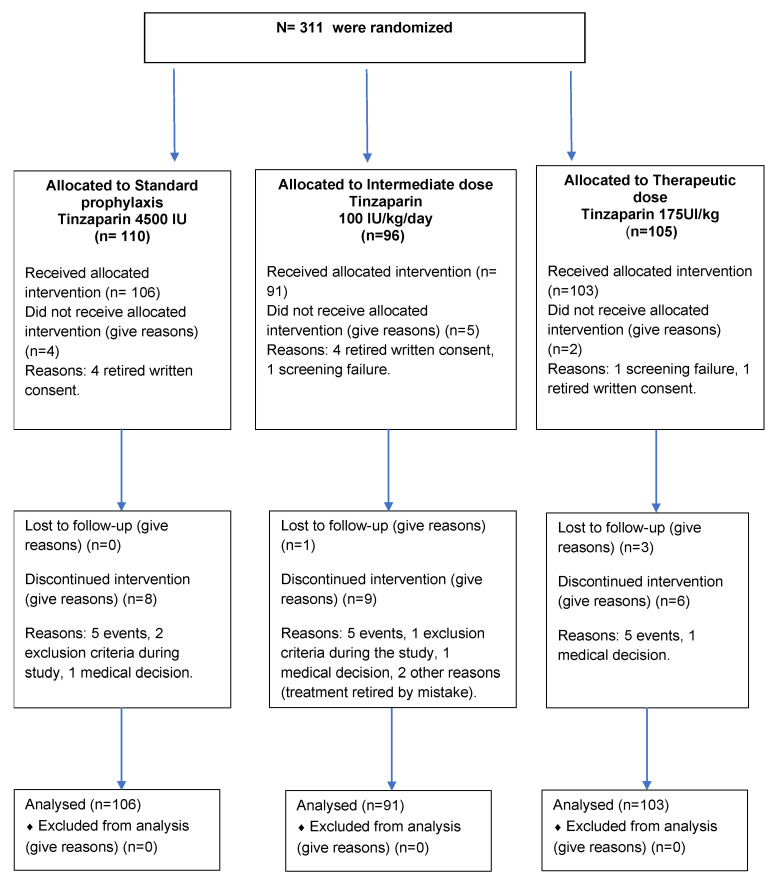
Flow chart for the PROTHROMCOVID trial.

**Figure 3 jcm-11-05632-f003:**
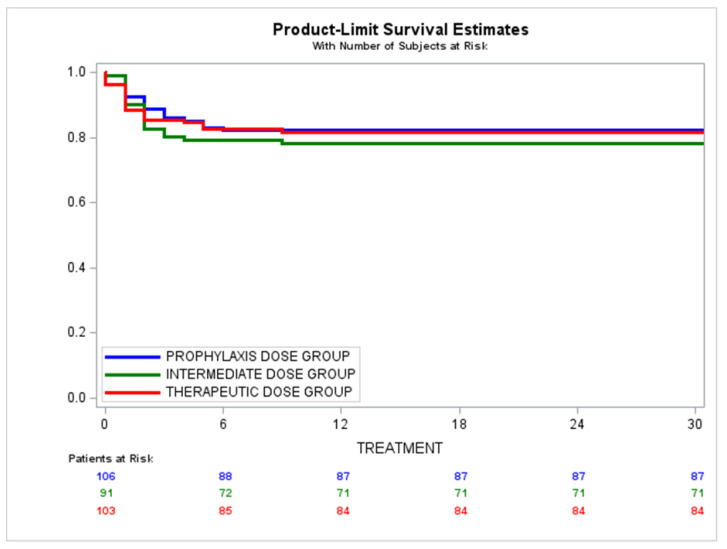
Overall survival of patient series, as per low molecular weight heparin group assignment at 30 days follow-up. Prophylactic dose group (tinzaparin 4500 IU/daily) Prob (95% CI): 0.82 (0.73, 0.88). Intermediate dose group (tinzaparin 100 IU/kg/day) Prob (95% CI): 0.78 (0.68, 0.85). Therapeutic dose group (tinzaparin 175 IU/kg/day) Prob (95% CI): 0.81 (0.73, 0.88). Log-rank test *p*-value = 0.75.

**Table 1 jcm-11-05632-t001:** Baseline characteristics of patients included in PROTHROMCOVID trial.

N = 300	Prophylaxis (4500 UI Tinzaparin) Group A (N = 106)	Intermediate (100 UI/kg Tinzaparin) Group B (N = 91)	Therapeutic (175 UI/kg Tinzaparin) Group C (N = 103)
Age, Mean (SD) (years) Weight, Median (Q1–Q3) (Kg) BMI Median (Q1–Q3)	54.1 (15.0) 79.6 (73.0–87.0) 28.5 (25.6–31.1)	56.5 (14.1) 78.5 (70.0–88.0) 28.6 (25.8–31.2)	58.5 (14.4) 78.9 (70.0–88.0) 28.7 (25.1–31.9)
Men, N (%) Women, N (%)	63 (59.4%) 43 (40.5%)	57 (62.6%) 34 (37.3%)	62 (60.2%) 41 (39.8%)
**Comorbidities**			
Hypertension, N (%)	29(27.4%)	34(37.4%)	36(34.9%)
Diabetes mellitus, N (%)	13 (12.3%)	17 (18.7%)	20 (19.4%)
Dyslipidemia, N (%)	26 (24.5%)	30 (33.0%)	36 (34.9%)
Smoking, N (%)	5 (4.7%)	6 (6.6%)	5 (4.8%)
Coronary heart disease, N (%)	4 (3.8%)	3 (3.3%)	3 (2.9%)
Chronic obstructive pulmonary disease, N (%)	3 (2.8%)	4 (4.4%)	5 (4.8%)
Chronic renal dysfunction, N (%)	1 (0.9%)	2 (2.2%)	3 (2.9%)
Prior stroke, N (%)	3 (2.8%)	1 (1.1%)	---%
Prior thromboembolic events, N (%)	1 (0.9%)	1 (1.1%)	2 (1.9%)
**Respiratory severity**			
Sat0_2_/ Fi0_2,_ Median (Q1–Q3)	353 (217–452)	346 (199–450)	342 (215–477)
**Laboratory test**			
Peak D-dimer, Median (Q1–Q3) (µg/dL)	618 (375–1100)	686 (404–1340)	620 (363–1200)
Platelets, Median (Q1–Q3) (×10^3^)	344 (269–436)	369 (299–439)	320 (246–401)
IL6 (Q1–Q3), Median (Q1–Q3) (mg/dL)	23.8 (7.8–50.1)	29.4 (5.7–63.8)	21.43 (7.4–43.9)
Creatinine, Median (Q1–Q3) (mg/dL)	0.76 (0.6–0.9)	0.73 (0.6–0.8)	0.71 (0.6–0.9)
Ferritin, Median (Q1–Q3) (ng/dL)	619 (274–1275)	775 (386–1347)	554 (271–1177)
CRP, Median (Q1–Q3) (mg/dL)	57.6 (25–107)	60.9 (14–142)	57.1 (27–131)
LDH, Median (Q1–Q3) (ng/dL)	336 (254–439)	333 (250–478)	301 (243–383)
ISTH-DIC score, Mean (SD)	2.42 (0.9)	2.56 (0.91)	2.33 (0.76)
**COVID-19 Treatment**			
Steroids, N (%)	94 (88.6%)	83 (91.2%)	91 (88.3%)
Remdesivir, N (%)	20 (18.8%)	16 (17.5%)	18 (17.4%)
Tocilizumab, N (%)	16 (15.1%)	18 (17.4%)	11 (10.6%)
**Vaccination Status**			
1 dose	8 (7.5%)	12 (13.1%)	12 (11.6%)
2 doses	12 (11.3%)	18 (19.7%)	16 (15.5%)

IL6 = Interleukin 6; CRP = C-reactive protein; ISTH-DIC score = International Society of Thrombosis and Haemostasis overt disseminated intravascular coagulation score.

**Table 2 jcm-11-05632-t002:** The primary endpoint was the composite outcome of death, need for mechanical ventilation (invasive or noninvasive or high-flow therapy via nasal cannula), and venous or arterial thrombosis within 30 days after randomization.

Primary Outcome	Prophylaxis Dose Tinzaparin 4500 IU/day (N = 106)	Intermediate Dose Tinzaparin 100 IU/kg/day (N = 91)	Therapeutic Dose Tinzaparin 175 UI/kg (N = 103)	Absolute Difference (* Intermediate Dose vs. Prophylactic Dose; ** Therapeutic Dose vs. Prophylactic Dose)	Risk Reduction (* Intermediate Dose vs. Prophylactic Dose; ** Therapeutic Dose vs. Prophylactic Dose)	*p*-Value
Primary endpoint (day + 30). N (%)	19 (17.9)	20 (22.0)	19 (18.4)	* 1 ** 0	* −4.0 (−7.2%, −15.3%) ** 0.5 (−9.9%, 10.9%)	0.769 ^1^
**Secondary outcomes**						
Death from any cause N (%)	2 (1.9)	3 (3.3)	2 (1.9)	* 1 ** 0	* 1.4% (−3.1%, 5.9%) ** 0.05% (−3.7%, 3.8%)	0.79 ^2^
Thrombotic event N (%)	4 (3.8)	2 (2.2)	2 (1.9)	* 2 ** 2	* 1.6% (−3.1%, 6.3%) ** 1.9% (−2.5%, 6.3%)	0.74 ^2^
ICU admission N (%)	7 (6.6)	6 (6.6)	10 (9.7)	* 1 ** 3	* 0.01% (−6.9%, 6.9%) ** −3.1% (−4.3%, 10.5%)	0.63 ^1^
High flow nasal cannula N (%)	13 (12.3)	14 (15.4)	13 (12.6)	* 1 ** 0	* −3.1% (−6.6%, 12.8%) ** 0.4% (−8.6%, 9.3%)	0.78 ^1^
Non invasive mechanical ventilation N (%)	4 (3.8)	4 (4.4)	2 (1.9)	* 0 ** 2	* −0.6% (−4.9%, 6.2%) ** 1.8% (−2.7%, 6.3%)	0.67 ^2^
Invasive ventilation N (%)	1 (0.9)	2 (2.2)	3 (2.9)	* 1 ** 2	* −1.2% (−2.3%, 4.8%) ** −1.9% (−1.8%, 5.7%)	0.60 ^2^
Progression WHO * scale, Median (Q1; Q3)	−0.43 (−1; 0)	0.13 (−0.5; 1)	0.06 (0; 1)	-	-	0.69 ^3^
Progression to adult respiratory distress syndrome by PaO_2_/FiO_2_ or SpO_2_/FiO_2_. N (%)	4 (3.8)	2 (2.2)	1 (1.0)	-	-	0.40 ^2^
Length of hospital stay, Median (Q1; Q3)	10.0 (6.0; 17.0)	9.5 (6.0; 24.0)	11.0 (6.0; 14.0)	-	-	0.96 ^4^
Major bleeding N (%)	-	-	-	-	-	-
Clinically relevant non major bleeding, N (%)	4 (3.8)	3 (3.3)	3 (2.9)	* 1 ** 1	* 0.5% (−4.7%, 5.6%) ** 0.9% (−4.0%, 5.7%)	1.00 ^2^

* Primary endpoint was composite outcome of death, intensive care unit admission, need for mechanical ventilation (invasive or noninvasive or high-flow therapy via nasal cannula), and venous or arterial thrombosis within 30 days after randomization. Secondary outcomes were measured at 90 days after randomization. ^1^ Chi-square test *p*-value. ^2^ Fisher’s exact test *p*-value. ^3^ Wilcoxon’s test *p*-value. ^4^ Kruskal–Wallis’ test *p*-value. ****** Therapeutic Dose vs. Prophylactic.

## Data Availability

Data supporting reported results can be shared under request at www.prothromcovid.org or nmunozr@salud.madrid.org.

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
