# Peer review of "Efficacy and Safety of Tinzaparin in Prophylactic, Intermediate and Therapeutic Doses in Non-Critically Ill Patients Hospitalized with COVID-19: The PROTHROMCOVID Randomized Controlled Trial"

_jcm, 2022, doi:10.3390/jcm11195632_

Round 1
Reviewer 1 Report
Line 77: acute respiratory distress syndrome (ARSD) acute respiratory distress syndrome (ARSD), the abbreviation needs to be corrected
Line 121: the need for full-dose anticoagulant therapy
Line 128: the sentence "Patients were screened at hospitalization" might be revised to patients were screened on admission.
Page 7 Table 2: Standard Prophylaxis Tinzaparine dosage unit should be corrected. (4500 IU/Day, not IU/KG)
Lines 246 to 248: The reported frequency of thrombotic events does not match the reported results in Figure 3 and should be clarified.
Line 259: Duplicated between should be corrected
Line 276: no benefit in survival to be revised to no survival benefit
Line 286: an early strategy of full-dose anticoagulant doses of heparin in non-critically ill; did the authors want to describe a plan switch after an early full-dose anticoagulant therapy?
Reviewer 2 Report
I read with interest the paper by Muñoz-Rivas et al. titled ”Optimal thromboprophylaxis strategies in non-critically ill patients with COVID-19 pneumonia. The PROTHROMCOVID 3 Randomized Controlled Trial”.
I thank the authors and the editor for the opportunity to review the manuscript.
Major points
The authors randomized 311 hospitalized patients with non-critical COVID-19 pneumonia 1:1:1 to low dose, intermediate and therapeutic dose tinzaparin. There was no difference between the three groups for the primary outcome that was a composite endpoint of symptomatic systemic thrombotic events, need for invasive or non-invasive mechanical ventilation, or death within 30 days. Further, there were no major bleeding and no difference between groups in regards to non-major bleeding.
The study is important and the manuscript is clearly written. However, there are some issues.
The study was unblinded and this fact is not mentioned in either the abstract or the methods section.
The study aimed for 200 patients in each group but the study was stopped at interim analysis due to futility and slow recruitment at approximately 100 patients in each of the three groups.
In their power calculation, the authors appears to have expected a high incidence of 24% for the primary outcome in the low dose group and a markedly lower risk at 12% in both the intermediate and high dose groups. It seems unlikely that the difference would be so large (no references are provided for these numbers) and it is unclear why the authors included both an intermediate and high dose group if they expected the incidence to be the same in the two groups.
The interim analysis and futility should be mentioned in the abstract.
The study was preregistered at Clinicaltrials.gov (NCT04730856) but here there is no mention of the planned number of patients nor that an interim analysis was planned.
In the Clinicaltrials.gov registration, the authors stated 6 primary outcome measures:
- Reduction of suspicion of systemic thrombotic symptomatic events
- Use of Mechanical ventilation
- Progression on the WHO Progression Scale during follow-up
- Overall survival at 30 days
- Length of hospital stay
- Length of ICU
These primary outcomes were changed to a single composite endpoint of symptomatic systemic thrombotic events, need for invasive or non-invasive mechanical ventilation, or death within 30 days. The composite outcome is preferable to the 6 primary outcomes but this change should be noted in the Limitations section.
Further, the secondary and safety outcomes were also modified.
There are a few issues with the presentation of data.
This therapeutic RCT appears not to have been registered at EDURA-CT.
There are few grammatical issues and I have commented on some of them below. Please revise, use a spell checker, use e.g. Grammarly or similar.
See specific points below.
Title
Consider a more informative title that explain the most important result.
Abstract
“did not appear to increase benefit regarding the likelihood of” – “did not appear to affect the risk of”.
Introduction
L77-79 – please add references.
“estimates more than that 18 million people died” – “estimates that more than 18 million people have died”.
”and need for ventilation” - ”and need for mechanical ventilation”
L85-87 and 87-89 – please add references.
“could improve the life expectancy” – please revise to e.g. outcome or mortality.
Consider to use either the term SARS-CoV-2 or COVID-19 rather than change between the terms.
“While the results have been more robust against higher than usual prophylactic doses” – the meaning is unclear, please revise.
L105 “patients with COVID-19 pneumonia” – please add “non-critical” or similar wording.
“in relation to developing complications of both major and minor bleeding” - “in relation to the risk of both major and minor bleeding”.
Methods
The manuscript should follow the CONSORT guideline as detailed by EQUATOR network and this should also be noted in the Methods section.
“The study design and full list of eligibility and exclusion criteria are provided in the supplementary material (Supplementary file eTable 1 and eTable 2).” – the inclusion and exclusion criteria appear to be included in Table A1 at the end of the manuscript whereas a table of the study design appears not to have been included in the manuscript.
Please state whether anticoagulation treatment was changed in case of e.g. a venous thromboembolic event, development of atrial fibrillation, etc.
“subcutaneously for seven days, after which thromboprophylaxis was maintained at the discretion of the investigator” – this seems strange, why continue treatment after 7 days after hospital discharge if there was no clear indication (e.g. previous VTE or AFIB or just according to clinical guidelines) and why was this decision taken by the investigator and not the attending physician.
”Secondary efficacy variables were” – please refer to Table A2.
“Secondary efficacy variables were the same endpoints at 30 and 90 days,” – this is unclear and does not seem to match the secondary outcomes reported in Table A2 (e.g. there is no thrombosis at 90 days and no mechanical ventilation mentioned in table A2). Please revise.
Table A2
“Composite endpoint including reduction in symptomatic” – delete “reduction”.
Please state the time frame(s) for each outcome.
“Safety: To determine the safety of the different prophylaxis and anticoagulation strategies in hospitalized patients with SARS-CoV-2 pneumonia.” – please clarify what the authors looked for, e.g. major and minor bleeding, adverse effects etc.
“(myocardial infarction, ischemic stroke, deep vein thrombosis, pulmonary thromboembolism confirmed with imaging tests)” – this is detailed for the secondary outcome, please use the same level of detail for the primary outcome.
Why is Orotracheal Intubation abbreviated as IOT?
“Progression of Acute Respiratory Distress Syndrome” – “Progression to Acute Respiratory Distress Syndrome”.
“Outcomes were adjudicated locally by the investigators” – please state whether only one person or at least two investigators had to adjudicate each case.
Please state the outcomes in the main text rather (e.g. using bullet points) than in a supplemental table.
Consider to describe the standard of care regarding use and doses or LMWH in the hospitals or briefly how it differed between the 18 centers – this is mostly important if patients had been treated for various days before inclusion in the study.
Statistical analysis
“Considering the main objective, the incidence in the control group was expected to be 24% and 12% in groups 2 and 3.” – What do the authors mean by control group and groups 2 and 3 – is it the low dose group, intermediate and high dose groups respectively? Please clarify.
“The sample size was calculated from a proportion of a 13% reduction in thrombosis in the control group and a difference of 5% compared to treatment groups.” – unclear, please revise.
Please add that per-protocol analyses were also carried out.
The authors argue that they use intention-to treat but that may not be the case as they only analyzed data from patients who received at least one dose of the allocated treatment. If these patients were excluded due to patient withdrawal of consent or screening failure this should be stated. If patients were excluded from the analysis because they for some other reason did not get or take the tinzaparin, it is not an intention-to-treat analysis.
Please describe the interim analysis, criteria cut-offs etc.
Results
Please omit the terms group A, B, and C as the just add confusion and call the groups the prophylaxis, intermediate and therapeutic dose group (like in table 1 and parts of the main text).
One decimal should be sufficient for the percentages.
”Flowchart in S figure 1” – this appears to be called Figure A1 at the end of the manuscript, please revise the names and references of all tables and figures.
”Received allocated intervention (n= 106)” – shouldn’t this be “Received at least one dose of the allocated intervention (n= 106)”?
Table 1
”Basal characteristics” – ”Baseline characteristics” or “Patient characteristics at baseline”.
“Age, Media (SD) (years) 54.1 (15)” – should be mean (SD). Please use the same number of decimals for all numbers for a single variable, please revise throughout the manuscript.
”Chronic renal dysfunction, N (%) ” – how was this defined, GFR < x?
Please report also Sat02 and Fi02 and not only the ratio.
Please add the prevalence of current and prior cancer and hematologic disease.
Hypertension, Diabetes mellitus and Dyslipidemia can be placed under comorbidities and smoking placed below gender, and “Cardiovascular risk factors” can be deleted.
Please report the prevalence of vaccinated patients in the table and clarify whether patients had received 1 or e.g. ≥2 doses.
*, † and ‡ can be omitted.
The P-values should be omitted in table 1 for the baseline characteristics but the values should be scrutinized for possible skewed distributions between the groups that may indicate problems in randomization.
Table 2
Please use a text table rather than an image.
Please replace “control” by “prophylaxis dose”.
“-7.2%, 15.3%” – should be “-7.2% -15.3%”.
“*1 **0” – this is unclear.
Exitus? Do the authors mean “Death from any cause”? Please clarify other instances also.
Please report the same secondary outcomes in the same order as in table 2A that shows the outcomes.
“Risk reduction 4.05%” should be -4.05% as the risk was higher in the intermediate as compared to the prophylaxis group. Please revise throughout the manuscript. One decimal is sufficient for percentages.
“In all three groups, most survived to medical discharge without the appearance of the primary endpoint.” – superfluous.
”group A, 95% CI: 0.82 (0.73-0.88);” please clarify whether these numbers refer to hospital discharge (as in the preceding sentence) or at 30 day follow-up.
”95% CI: 0.82 (0.73-0.88)” – ”0.82 (95% CI: 0.73-0.88)”.
Figure 2
Please write “Composite endpoint” and write out IMV etc.
”No differences were observed in survival” – do the author mean “event-free survival in regards to the primary outcome”?
“In terms of safety, the rate of bleeding was very low in all 3 groups.” – Delete, this is clear from the next sentence.
Table A3 – the row “total” can be omitted. Consider to delete the table.
Figures 2 and 3 would likely be more readable as tables like Table 2 in which P-values, tests etc. can also be reported.
Also, why report the incidence of each outcome as a percentage of the patients with an outcome? What about patients with more than one outcome? It would be much more informative to see the number and percentage of the group population as in table 2.
ARDS and High flow have almost the same color.
“4 patients in the prophylaxis group (3.8%)” - the exact numbers should be presented in a table and not repeated in the main text whereas the main text should briefly describe the results.
“The World Health Organization (WHO) progression scale indicated no intergroup” – should also be included in the table with the secondary outcomes.
”no evidences” – ”no evidence”.
The authors mention per-protocol analysis but do not report whether the results were similar as the intention-to-treat analysis.
Discussion
“treatment with tinzaparin in relation to prophylactic, intermediate, or therapeutic doses in relation to the probability” - “treatment with prophylactic, intermediate, or therapeutic dose tinzaparin in relation to the risk of the composite outcome of”.
L267 “in patients with COVID-19 pneumonia.” – please add that patients were hospitalized and not critically ill.
“In this regard, the results of our trial provide evidence about the use of LMWH in non-critical patients with pneumonia due to COVID-19” – This seems superfluous and here it is unclear what can be learned from the study, please revise and expand on what is learned from the trial, e.g. that there seems to be no advantage of a higher dose although the risk of major bleeding appears to be low regardless of dose.
“and the high rate” – try e.g. “faced with the high rate”.
”organ-free support days” – unclear, please revise.
”than the one observed” – ”than those observed”.
“with the first dose of tinzaparin administered within the first 24 hours” – Were all patients recruited within the first 24 hours? If so, this should be made clear in the methods or results section.
“against intermediate-to-therapeutic anticoagulation” – unclear, do the authors mean “as compared to” or “rather than”?
“The PROTHROMCOVID study confirms the non-superiority of intermediate doses” – add “and therapeutic”.
“suggests that this strategy should be abandoned” – consider to clarify that use of intermediate and therapeutic doses likely should be abandoned in this patient group.
Regarding ref 21, are there data on a different effect in hematologic patients? If there are a significant number of oncologic and hematologic patients in the present study, this topic should be discussed more in detail.
”the option of anticoagulation” – unclear, do the authors mean “therapeutic”?
”unsafe major bleeding” – delete ”unsafe”.
L350-361 from “At this point there were 19“ – this should be described in the methods/results section as appropriate and should only be mentioned here in breif.
”influenced the favorable efficacy” – delete. The efficacy was similar in the groups and the effect of LMWH can not be evaluated by the study as there was no placebo group.
Conclusion
“in the likelihood of thrombotic event, ” – “on the risk of thrombotic events, use of”.
References
The DOI of published articles is unnecessary. Please see the journal policy
I have no conflicts of interest to declare.
Author Response
Please see the attachment , thank you very much.

Reviewer 3 Report
1. General comments
This study investigated appropriate doses of anticoagulants, tinzaparin, among patients with non-critical COVID-19. The study design is a multicenter randomized controlled trial followed by the CONSORT 2010 statement. The author described that the dose of tinzaparin did not appear to benefit the thrombotic event or exacerbation. The result of this study provided important information about anticoagulant therapy for patients with COVID-19. However, this study has not reached the estimated sample size and did not achieve a significant difference.
2. Specific comments
Major comment
a) In some countries, tinzaparin was not approved, and enoxaparin was more common as Low-molecular-weight heparin (LMWH). I recommend that the author add a basic explanation about tinzaparin and the evidence for preventing thrombosis in patients with COVID-19.
b) The author should describe a protocol for assessing and diagnosing thrombosis, for example, routine assessment by venous ultrasonography or symptom-based survey by contrast-enhanced computed tomography. Additionally, the author should add detail about thromboses, such as pulmonary embolism, deep vein thrombosis, and arterial thrombosis.
c) Figure 3 shows the percentage of the frequency of the main secondary efficacy endpoint. First, the actual number and frequency of the figure and those in the text were different. Second, there is supposed to be duplicated among the events. The author should change the figure to match the content of the main text.
Minor comment
a) The appearance (space between lines and number of decimal places (ISTH-DIC, Tocilizumab)) of Table 1 is inconsistent. The author should revise it.
b) ISTH-DIC score was only described as mean and SD in table 1. Did it mean that this variable had normality?
c) The author should add “(Q1-3)” behind “CRP Median” in table 1.

Round 2
Reviewer 2 Report
I commend the authors for their revision, the manuscript is improved.
Some minor issues remain.
Major points
The authors argue that they use an intention to treat analysis – yet they only included patients in the analysis who received at least one dose of the allocated tinzaparin. The intention to treat analysis should include all patients who were randomized regardless of whether the allocated dose was administered. I raised this point in my first review but the authors have made no correction.
The authors should include all randomized subjects, and if not, they should clarify in the methods section that a modified intention to treat analysis was used as the analysis only included patients who received at least one dose. For the safety outcomes, the modified ITT analysis may be more appropriate (they can not be expected to bleed if they did not get the medication).
When reviewing the flow chart in figure 2 it seems that all patients that were included actually did get at least one dose while the drop outs were due to withdrawal of consent or screening failure and not because they did not get a dose (meaning that all this may be a non-issue?). Thus the wording “received at least one dose” might be omitted in the methods and in the results section the authors can note that x subjects were excluded from the analysis due to withdrawal of consent or screening failure while all other patients did receive at least one dose and were all included in the analysis.
Minor comments
Abstract
Please add that all tinzaparin doses are once daily.
Introduction
“could improve outcomes events” – “could improve clinical outcome”.
Methods
The authors included patients with COVID-19 pneumonia, but the definition of pneumonia seems to be missing, did patients have to have pneumonic consolidations/infiltrations on chest x-ray or CT scans?
“(Supplementary file eTable 1 and eTable 2)” – these tables at the end of the manuscript are called “Table A1.” And “Table A2.” not eTable?
“Weight between 50 and 90 kg” in the table and 50-100 kg in the manuscript.
”Platelet disease <80,000/μl.” – ” Platelet count <80,000/μl.”
“Kidney failure with MDRD <30 ml / min” – “MDRD”? do the authors mean estimated GFR?
Table A2 – each abbreviation is stated only once and should be omitted.
In figure 1 the part “Stratified by age, HTA (hypertension?), sex” is not explained in the main text.
The resolution of the figure is low. The inclusion criteria is placed on top of the follow up – please move or omit.
”tinzaparin 136 administered” – ”tinzaparin was 136 administered”.
“patients could receive prophylactic LWMH” – this is contradicted by the statement on dose escalation. Maybe add “prophylactic or higher dose LWMH”.
“Patients received anticoagulation treatment when they developed a thromboembolic event” – try e.g. “Patients were changed to therapeutic dose anticoagulation treatment if they developed a thromboembolic event”.
L152 “Outcomes”
Here there is no reference to Table A2 which contain the same information. Consider to add the secondary outcomes 8 to 12 to the main text and delete Table A2 as it is a repetition.
Similarly the supplemental Table A1 can be omitted and as the inclusion and exclusion criteria can be stated in the main text.
”clinical suspection” – ” clinical suspicion”?
Please add the Edura-CT registration number.
Results
Fig 2: some text is missing.
“Age, Mean (SD) (years) 54.1 (15)” and “BMI Median (Q1-Q3) 28.5 (25-31)” – please revise the decimals throughout the manuscript, e.g. “54.1 (15.x)” and “28.5 (25.x-31.x).
Please clarify in the Methods section whether the e.g. CRP in table 1 is the peak value or the value upon hospital admission. For D-dimer table 1 states that it is the peak.
”(17.1 %) in prophylactic” – ”(17.1 %) in the prophylactic” – please revise throughout as applicable. Articles are missing in many sentences.
Table 2 “Risk reduction (*Intermediate dose vs. Control; ” please use the term “prophylactic” rather than “control” as in the rest of the manuscript.
”2 (1.89)” and “4.05 (-7.2%, " – as I stated in my previous review, one decimal is sufficient for percentages.
L336-9 is a repetition of L292-4.
Please add that there were one oncological patient in each group and that no patients had hematological diseases.
Please add that the results of a per-protocol analysis were similar to the intention-to-treat analysis.
Discussion
“the non-superiority of intermediate doses 412 and therapeutic” – “the non-superiority of intermediate 412 and therapeutic doses”.
“prophylactic intensity over interme-417 diate or therapeutic intensity” – replace “intensity” by “dose”.
L439-42: can be omitted.
“tinzaparin, such as enoxaparin or 453 bemiparin,” – tinzaparin and enoxaparin are different drugs, do the authors mean “LMWH such as tinzaparin, enoxaparin or bemiparin”.
“reported after the 455 use of tinzaparin” – “reported on the use of tinzaparin”.
“expected to 461 be milder due to” – consider to add “less pathogenic variants of COVID-19”.
Conclusion
Please add that the patients had COVID pneumonia.
”However, administration of full or intermediate heparin doses in 467 these patients is safe” – there were no major bleeding in the 100/100 patients in the intermediate and high doses groups, yet the literature does inform us that bleeding occur and the results may be due to chance. Please revise the wording, e.g.
“However, the risk of bleeding related to intermediate or full heparin doses appears to be low in these patients.”
L538: “and the null hypothesis of equality of time to even between the three 538 treatment groups cannot be rejected” – unclear.
Author Response
Please see the attachment, I also would like to say that information has been modified at clinialtrials.gov. Thank you

Reviewer 3 Report
Minor revision
・page 9, line 293
The percentage of the primary endpoint in the prophylaxis group is incorrect, 17.1% → 17.9%.
・Table 1
"ISTH-DIC score, Mean SD)" is strange. The author should correct this description appropriately.
・Table 2
"Progression WHO scale, Mean (Q1-Q3)" is a strange description. The author should correct it (e.g. Mean (SD) or Median (Q1-Q3)).
Author Response
Please see the attachement, thank you.
